# Trehalose metabolism confers developmental robustness and stability in *Drosophila* by regulating glucose homeostasis

Ryota Matsushita[1,2] & Takashi Nishimura[1,2 ✉]

Organisms have evolved molecular mechanisms to ensure consistent and invariant phenotypes in the face of environmental fluctuations. Developmental homeostasis is determined by two factors: robustness, which buffers against environmental variations; and developmental stability, which buffers against intrinsic random variations. However, our understanding of these noise-buffering mechanisms remains incomplete. Here, we showed that appropriate glycemic control confers developmental homeostasis in the fruit fly *Drosophila*. We found that circulating glucose levels are buffered by trehalose metabolism, which acts as a glucose sink in circulation. Furthermore, mutations in trehalose synthesis enzyme (*Tps1*) increased the among-individual and within-individual variations in wing size. Whereas wild-type flies were largely resistant to changes in dietary carbohydrate and protein levels, *Tps1* mutants experienced significant disruptions in developmental homeostasis in response to dietary stress. These results demonstrate that glucose homeostasis against dietary stress is crucial for developmental homeostasis.

[1] Laboratory for Growth Control Signaling, RIKEN Center for Biosystems Dynamics Research (BDR), 2-2-3 Minatojima-Minamimachi, Chuo-ku, Kobe, Hyogo 650-0047, Japan. [2] Graduate School of Biological Science, Nara Institute of Science and Technology, 8916-5 Takayama, Ikoma, Nara 630-0101, Japan. ✉email: takashi.nishimura@riken.jp

Organisms are acutely and chronically exposed to various environmental conditions, such as temperature, humidity, radiation, and nutrition. Despite such perturbations, many organisms can develop relatively consistent phenotypes, due to their intrinsic robustness against environmental changes. The concept of developmental homeostasis refers to the ability of an organism to maintain phenotypic consistency and reproducible outcomes in response to genetic and environmental variations[1].

Phenotypic variability during developmental processes is determined by two critical components: robustness (including synonymous terms, such as canalization) and developmental stability. Developmental robustness, or microenvironmental canalization, buffers phenotypic variations against genetic and/or environmental conditions[2]. Developmental stability buffers against local and small, random perturbations, termed developmental noise[3,4]. Developmental robustness is commonly evaluated as the phenotypic variations among individuals that share genetic and environmental conditions. In contrast, the degree of developmental stability is evaluated by using fluctuating asymmetry (FA), which is the measure of random deviations from perfect symmetry. Lower inter-individual variation (IIV) for a given trait reflects robustness, whereas reduced FA reflects the capacity of an organism to buffer against stochastic noise generated during development. Although the heritability of FA is typically very low, suggesting that FA has no or very little additive genetic basis, increased developmental stability has been associated with increased fitness[5,6].

The genetic basis of developmental homeostasis has been intensively studied using the wings of the fruit fly *Drosophila melanogaster* as a model system[7,8]. Both robustness and developmental stability have been considered to be polygenic traits whose variation is primarily determined by many genes with small impacts, instead of by few genes with substantial impacts[9,10]. In *Drosophila*, phenotypic variations can be assessed among genotypically identical individuals grown under controlled nutrient conditions. However, even under controlled conditions, phenotypic variations in bilaterally symmetrical structures can occur, both among and within individuals. The causes of within-individual variations (e.g., FA) are thought to be associated with the stochastic nature of developmental noise because the left and right structures are generated under the same external and internal environmental conditions, using the same genotypes.

Robust body sizes and symmetry require highly connected genetic networks, which stabilize the precise regulation of cellular growth and proliferation during development[7,8]. The growth of developing organs is regulated by nutrient-sensitive systemic signals and local signals, such as morphogen gradients[11,12]. Thus, appropriate control of metabolic homeostasis is crucial for post-embryonic development in response to dietary fluctuations. Metabolic homeostasis is maintained by highly regulated and intricate feedback systems that ensure the delicate balance among food intake, energy expenditure, and resource storage[13,14]. The evolutionarily conserved insulin-like peptide is the primary nutrient-sensitive anabolic hormone, which promotes the absorption of postprandial blood glucose into several organs and regulates tissue growth[15,16]. In *Drosophila*, aerobic glycolysis is indispensable for body growth during larval development[17,18], suggesting that glycemic control is a central component for the maintenance of developmental integrity by providing a constant supply of energy to each organ.

In *Drosophila*, the dominant hemolymph sugar is the disaccharide trehalose, which acts as a glucose sink in circulation[19,20]. Trehalose is synthesized by trehalose-6-phosphate synthase 1 (Tps1) in the fat body, a *Drosophila* organ analogous to the mammalian liver. Larvae deficient in trehalose metabolism show developmental delays and growth defects[21]. Moreover, larvae with high-sugar- or high-fat-diet-induced obesity have been reported to manifest metabolic syndromes, such as increased circulating glucose levels, higher fat content, and insulin resistance, and exhibit severe growth retardation[22,23]. Thus, in addition to nutrient-sensitive systemic signaling networks, the homeostatic control of blood glucose is vital for average body growth during development. Despite the importance of glycemic control for metabolic homeostasis, the mechanisms that govern the interplay between glucose homeostasis and developmental homeostasis remain unknown.

Here, we report that trehalose metabolism acts as a metabolic buffer that sustains robust development in *Drosophila melanogaster*. We found that *Tps1* mutants showed feeding-associated hyperglycemia and fasting hypoglycemia and consequently exhibited vulnerability to metabolic perturbations. We further revealed that *Tps1* mutants showed increased IIV and FA for adult wing size. Notably, the defects on FA in *Tps1* mutants were sharply exacerbated by a low-glucose (LG) diet, whereas a high-glucose (HG) diet attenuated the mutant phenotype. However, both HG and low-protein diets worsened the IIV among *Tps1* mutants, indicating that environmental changes differentially affected developmental robustness and stability. In conclusion, the present study provides direct evidence that glucose homeostasis impacts developmental homeostasis. Trehalose metabolism likely evolved to maximize developmental homeostasis by buffering glucose fluctuations in response to environmental variations.

## Results

**Trehalose metabolism functions in glucose homeostasis.** To understand the significance of glucose homeostasis on developmental homeostasis, we utilized reverse genetics, which facilitates the direct manipulation of circulating sugar levels (Fig. 1a). To this end, we made use of a hypomorphic *Tps1* allele, named *Tps1*[MI03087], because null mutations in *Tps1* result in complete lethality during the pupal stage[21]. A *Minos* transposon was inserted into the first intron of *Tps1*, just before the translational start codon (Fig. 1b), which reduces the mRNA expression level to ~20% of the wild-type level in mid-third-instar larvae[21]. We hereafter refer to the *Tps1*[MI03087] line backcrossed with a control strain $w^-$ as *Tps1*[MIC]. The metabolic analysis revealed that the trehalose levels in homozygous *Tps1*[MIC] mutants were reduced to 20% of those in control at the post-feeding wandering stage (Fig. 1c). Glucose, glycogen, and triglyceride (TAG) levels did not change at the level of whole larvae.

Glucose levels fluctuate more dynamically than trehalose levels in the circulating hemolymph in response to dietary challenges[24,25]. To investigate the physiological role of trehalose metabolism in glucose homeostasis during the feeding stage, we determined the absolute concentrations of circulating sugars in mid-third-instar larvae, using liquid chromatography coupled with tandem mass spectrometry (LC-MS/MS). We found that the glucose levels in *Tps1* mutants were higher than those in control larvae when fed with a normal diet (ND), containing 10% glucose (Fig. 1d). The increased glucose levels in *Tps1* mutants could be reversed by chronic feeding with a diet devoid of glucose (LG diet). In contrast, HG diet containing 20% glucose further increased glucose levels in both control and *Tps1* mutant larvae, indicating that circulating glucose levels reflect dietary glucose levels. *Tps1* mutants also showed increased levels of the polyol pathway metabolites, sorbitol and fructose (Fig. 1a), whose levels largely corresponded with glucose levels. The diet-dependent changes in trehalose levels were less pronounced than those observed for glucose and fructose.

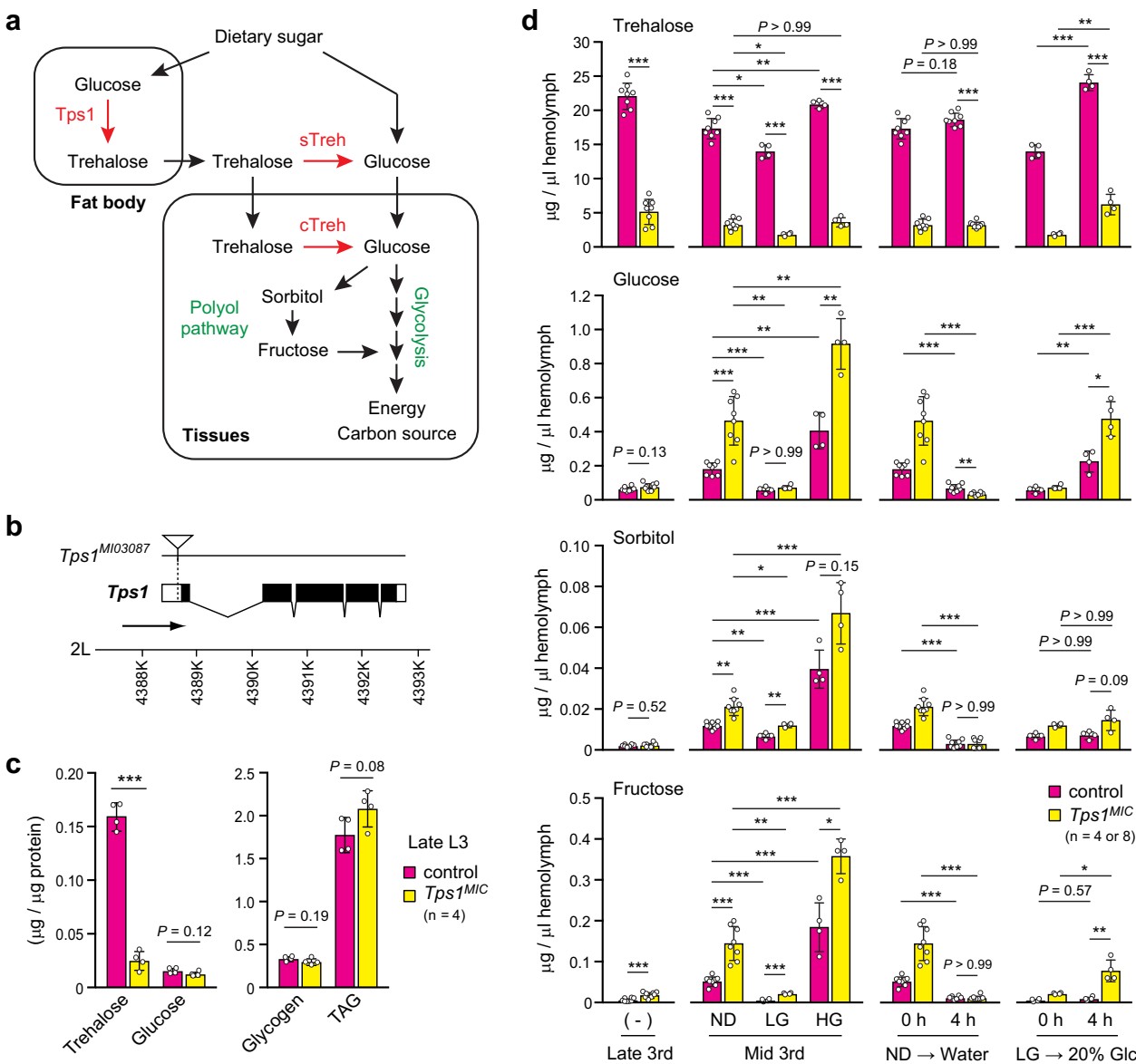

**Fig. 1 Trehalose metabolism functions in glucose homeostasis. a** Overview of trehalose and glucose metabolism. Proteins that function in trehalose metabolism are shown in red. **b** Schematic representation of the *Tps1* locus. Protein-coding regions and untranslated regions are represented by black boxes and white boxes, respectively. The *Minos* insertion site is marked with an inverted triangle. **c** Amounts of trehalose, glucose, glycogen, and TAG in *Tps1MIC* mutants at the wandering stage. **d** Circulating sugar levels under various dietary conditions. ND, normal diet; LG, low-glucose diet; HG, high-glucose diet; Glc, glucose. Composition of diets is shown in Fig. 5a. *$p < 0.05$, **$p < 0.01$, ***$p < 0.001$; unpaired two-tailed Student's *t*-test (**c**), unpaired two-tailed Student's *t*-test with Bonferroni correction (**d**). Results are presented as the mean ± SD. The numbers indicate the number of biological replicates.

To further examine the changes induced by acute challenges, we transferred mid-third-instar larvae grown on ND to a water-only diet for 4 h. The acute deprivation of diet reduced circulating glucose levels, but not trehalose levels, in control larvae (Fig. 1d). Interestingly, *Tps1MIC* mutants showed lower glucose levels than control larvae, indicating that *Tps1* mutants experienced post-prandial hyperglycemia and fasting hypoglycemia. Moreover, acute high-glucose challenges (from LG to 20% glucose diet for 4 h) increased glucose and fructose levels more drastically in *Tps1* mutants than in control larvae. Because trehalose is the dominant hemolymph sugar, with concentrations more than 100-fold those for glucose (Fig. 1d)[25,26], we suggest that trehalose metabolism buffers the fluctuations of circulating glucose levels in response to acute or chronic dietary challenges, playing a crucial role in glucose homeostasis.

**Trehalose catabolism cell-autonomously regulates organ growth.** We next investigated the consequences of impaired trehalose metabolism on organ growth. Although the underlying mechanisms remain to be clarified, sexual dimorphism for survival rate was observed among *Tps1MIC* mutants: approximately 30% of males and 70% of females were eclosed under well-nourished ND conditions[21]. The overall morphology of eclosed mutant flies was indistinguishable from those of control flies in both males and females (Fig. 2a). Detailed analyses revealed that the wing areas of *Tps1MIC* mutants were smaller than those of heterozygous flies in both males and females (Fig. 2b), indicating that the chronic reduction in trehalose levels decreases organ size.

Organ size variations result from changes in cell sizes and/or cell numbers. To determine whether the reductions in wing size observed in *Tps1MIC* mutants were caused by reduced cell

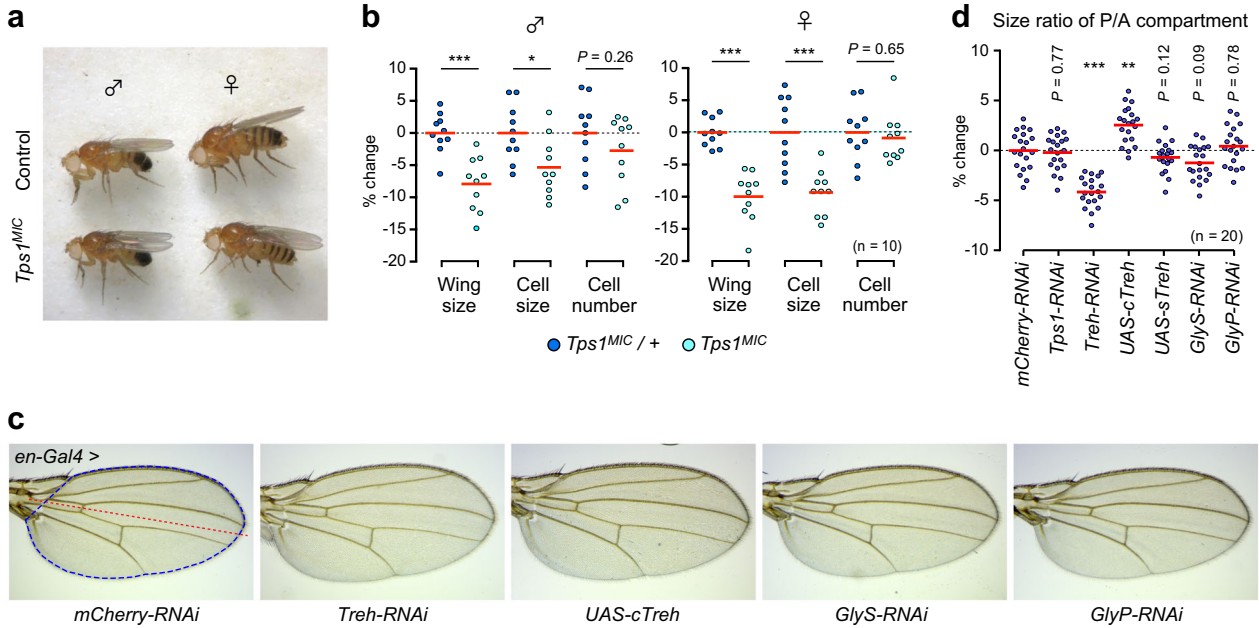

**Fig. 2 Trehalose catabolism cell-autonomously regulates organ growth. a** $Tps1^{MIC}$ mutant adults are morphologically normal. **b** Fold changes in wing size, cell size, and cell number are shown relative to heterozygous mutants. **c** Representative images of adult wing. *en-Gal4* was used to drive UAS transgenes in the posterior region of the wing. The red line indicates the boundary between the posterior and anterior compartments. The blue encircled region was measured to obtain the wing size. **d** The size ratio between the posterior/anterior (P/A) compartment areas in the adult wing. Fold changes are shown relative to control (*en-Gal4 > mCherry-RNAi*). *$p < 0.05$, **$p < 0.01$, ***$p < 0.001$; unpaired two-tailed Student's *t*-test (**b**), one-way ANOVA with Dunnett's post hoc test (**d**). The numbers of flies analyzed are indicated.

numbers, cell sizes, or both, we determined the wing hair density. The adult wing is composed of a two-layered epithelial sheet. Because each cell in the wing generates a single hair, hair density is inversely correlated with cell size. The hair density increased in *Tps1* mutants compared with that in heterozygous flies in both males and females, whereas the calculated cell numbers did not change (Fig. 2b). Thus, $Tps1^{MIC}$ mutant wings are small because they contain smaller cells with minimal changes in cell numbers.

We next investigated the cell-autonomous contributions of trehalose catabolism to organ growth using tissue-specific genetic manipulations. The gene locus of the trehalose hydrolysis enzyme *trehalase* (*Treh*) produces two types of proteins with distinct localizations through alternative splicing: cytoplasmic Treh (cTreh) and secreted Treh (sTreh) (Fig. 1a). *cTreh* and *sTreh* are expressed in various tissues during development and are functionally redundant[21,27]. The knockdown of *Treh* at the posterior compartment of wing imaginal discs using *en-Gal4* did not affect the overall wing morphology of adults (Fig. 2c). However, the knockdown of *Treh* significantly reduced the size of the posterior region compared with the size of the anterior region (Fig. 2d). In contrast, the overexpression of *cTreh*, but not of *sTreh*, increased the size of the posterior region. The knockdown of *Tps1* in wing discs had no effect on wing size because *Tps1* is exclusively expressed in the fat body[21] and, thus, served as an additional negative control. These results suggest that trehalose hydrolysis affects organ growth in a cell-autonomous manner.

In addition to trehalose, branched polysaccharide glycogen functions as a form of intracellular glucose storage. Larvae deficient in glycogen metabolism manifest various metabolic alterations and growth defects[28]. However, the wing morphologies and sizes in *glycogen synthase* (*GlyS*) and *glycogen phosphorylase* (*GlyP*) knockdown adults were indistinguishable from those in control flies (Fig. 2c, d), suggesting that glycogen metabolism is dispensable for wing development and morphogenesis under normal growth conditions.

**Defects in trehalose metabolism reduce developmental homeostasis and fitness.** We next investigated developmental homeostasis in *Tps1* mutant flies. We examined two components: size variation among individuals (IIV), as a measure of developmental robustness/canalization, and asymmetric size variation within an individual (FA), as a measure of developmental stability. We found that $Tps1^{MIC}$ mutants showed significantly increased IIV and FA for both sexes compared with genetically matched $w^-$ control and $Tps1^{MIC}$ heterozygous flies (Fig. 3a–c). Moreover, these changes were canceled by the presence of a single *Tps1* genomic rescue construct. These results indicate that homozygous mutations in *Tps1* reduce both developmental robustness and stability.

Although FA is not necessarily a general bioindicator of fitness[5,6], FA in males has been linked to sexual selection in *Drosophila*[29,30]. Therefore, we next tested the reproductive fitness of *Tps1* mutant males by a mating competition assay, in which we crossed a virgin female from a wild-type strain, either *Canton S* or *Oregon R*, with age-matched $w^-$ control and *Tps1* mutant males and then determined the genotypes of the offspring, to identify which males mated successfully. This assay revealed that *Tps1* mutant males had a lower mating success rate than the control males (Fig. 3d). In the absence of any competitor, ~60% of *Tps1* mutant males generated offspring, with a reasonable number of progenies (Fig. 3e). Thus, more than half of the mutant males were fertile, although a noticeable population of males was sterile. Although a causal relationship between reduced developmental homeostasis and reproductive ability remains unclear, these results reveal that *Tps1* mutant males show reduction in both reproductive fitness and developmental homeostasis.

In addition to acting as a reserve for glucose homeostasis, circulating trehalose plays potential roles in various biological processes, such as the maintenance of water homeostasis and the regulation of autophagy[27,31]. Moreover, Tps1 can prevent apoptosis in yeast by acting as a "moonlighting" protein[32]. To

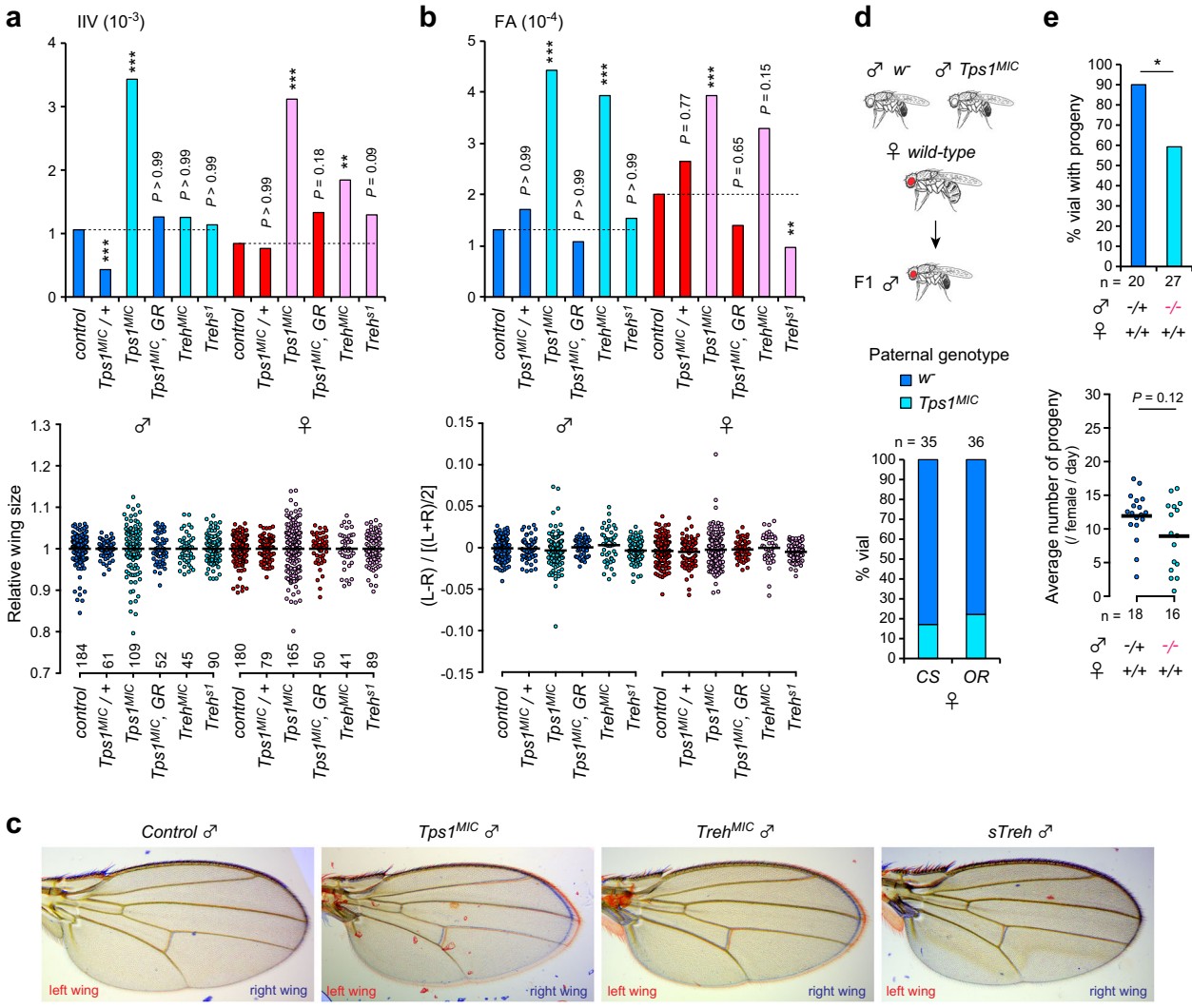

**Fig. 3 Defects in trehalose metabolism reduce developmental homeostasis and fitness. a, b** Inter-individual variation (IIV) (**a**) and fluctuating asymmetry (FA) (**b**) in wing size for each genotype and for each sex. The upper graphs indicate the variance of wing sizes (**a**) and size-corrected FA (**b**). The lower dot plots indicate the distributions of relative wing sizes normalized against the average size among individuals (**a**) and the distribution of wing size variations within each individual (**b**). L, left-wing size; R, right-wing size. GR indicates *Tps1* genomic rescue construct. **c** Merged images of left (red) and right (blue) wings from control, *Tps1*, and *Treh* mutant males. **d** *Tps1^{MIC}* mutant males show lower reproductive success rates compared with control males, as revealed by a mating competition assay. Paternal genotypes were checked in F1 males that derived from a female crossed with two males. *CS*, Canton S; *OR*, Oregon R. **e** Percentage of vials with progenies and the average number of progenies. More than half of the *Tps1^{MIC}* mutant males are fertile and produce a normal number of progenies. −/−, *Tps1^{MIC}* mutants; +/+, control *w^-*. *$p < 0.05$, **$p < 0.01$, ***$p < 0.001$; F-test with Bonferroni correction (**a, b**), Fisher's exact test, Mann–Whitney $U$ test (**e**). The numbers of flies analyzed are indicated.

determine whether the reduced developmental homeostasis observed in *Tps1* mutants was caused by decreased trehalose/Tps1 levels or by defects in sugar homeostasis, we analyzed hypomorphic *Treh* mutants because null mutations in *Treh* resulted in complete lethality during the pupal stage[27], precluding any analyses of these mutations in adults. We found that homozygous mutations in *cTreh* (*Treh^{MIC}*) showed increased FA in males, similar to the phenotypes observed in *Tps1^{MIC}* mutants (Fig. 3b, c). Moreover, *Treh^{MIC}* mutants showed increased IIV in females (Fig. 3a). In contrast, homozygous mutations in *sTreh* (*Treh^{s1}*) resulted in no increases in either IIV or FA. Thus, the wing phenotype observed in *Tps1* mutants was partially recapitulated by *Treh* mutant flies, suggesting that appropriate glucose homeostasis during the growth phase plays a role in developmental robustness and stability.

**Phenotypic specificity and severity on developmental homeostasis by perturbations of trehalose metabolism.** To further examine the specificity of genetic mutations involved in trehalose metabolism on developmental homeostasis, we analyzed two wild-type strains and compared them with a control strain, *w^-*, which we used as a genetically matched control. We found that Canton S and Oregon R showed almost equivalent levels of IIV and FA to those observed for the control *w^-* in both sexes (Fig. 4a, b), indicating that IIV and FA are relatively stable in at least these three genotypes. We next investigated the contribution of glycogen metabolism to developmental homeostasis using *GlyS* and *GlyP* null mutants[28]. *GlyS* mutants slightly increased IIV in females but did not result in increased FA (Fig. 4a, b). In contrast, *GlyP* mutants displayed no changes in either IIV or FA compared with control flies. These results

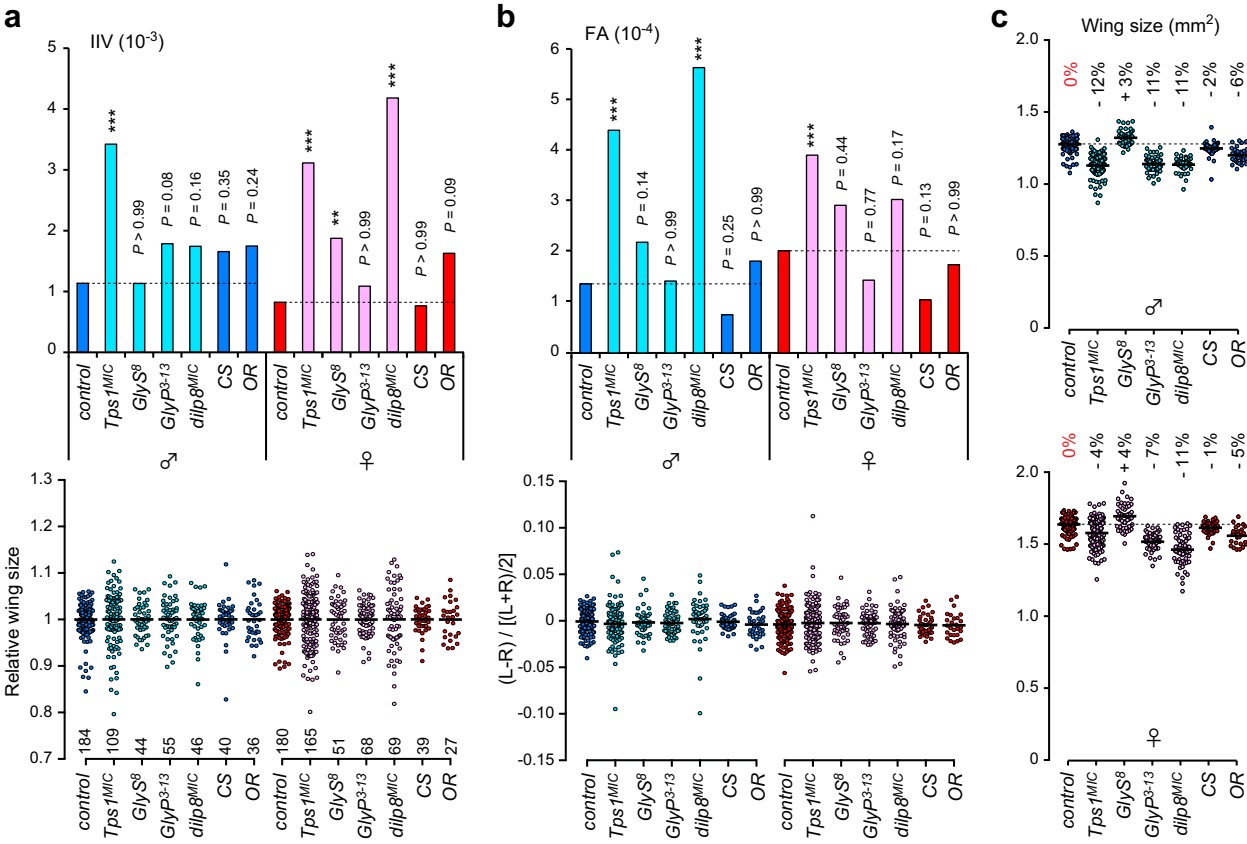

**Fig. 4 Phenotypic specificity and severity on developmental homeostasis by perturbations of trehalose metabolism. a, b** Inter-individual variation (IIV) (**a**) and fluctuating asymmetry (FA) (**b**) in wing sizes for each genotype and for each sex. The upper graphs indicate the variance of wing sizes (**a**) and size-corrected FA (**b**). The lower dot plots indicate the distributions of relative wing sizes normalized against the average size among individuals (**a**) and the distribution of wing size variations within each individual (**b**). Control, $w^-$; CS, Canton S; OR, Oregon R. All mutants were assessed on a $w^-$ background. **c** Wing size for each genotype and for each sex. Dotted lines indicate the average values for the control males and females. The percentage differences in the average values, relative to control values, are shown in the graph. $**p < 0.01$, $***p < 0.001$; F-test with Bonferroni correction. The numbers of flies analyzed are indicated.

indicate that the contribution of glycogen metabolism to developmental homeostasis is limited.

We noticed that the wing area in GlyP mutants was smaller than that in control flies for both males and females (Fig. 4c). Because GlyP mutants are smaller than controls, as assessed by both pupal volume and adult weight[28], the size reductions observed in GlyP mutant wings likely occur in a non-cell-autonomous manner. $Tps1^{MIC}$ mutant flies also had smaller wings (Fig. 2b), and the degrees of reduction were comparable to those observed in GlyP mutants. These results imply that the reduction in organ size is unrelated to the reduction in developmental robustness and stability.

We further examined the phenotypic severity of the size deviations observed in the Tps1 mutant flies. To this end, we re-evaluated dilp8 mutants, which demonstrated reduced developmental robustness and stability[33]. Dilp8 is a damage-inducible peptide secreted from imaginal discs and is a strong candidate for the long-sought hormone that stabilizes growth across the body and ensures robust symmetry and proportionality[8]. Consistent with previous reports, we observed increased size deviations in $dilp8^{MIC}$ mutants, although sex differences were also detected in our experimental conditions; $dilp8^{MIC}$ mutants significantly increased IIV in females but not in males, whereas these mutants increased FA in males but not in females (Fig. 4a, b). The reductions in developmental robustness and stability observed in

Tps1 mutants are approximately comparable to those observed in dilp8 mutants, further supporting the conclusion that trehalose metabolism is crucial for developmental homeostasis.

**Dietary conditions strongly influence developmental homeostasis in Tps1 mutants.** We next examined the impacts of dietary conditions during the larval period on developmental homeostasis (Fig. 5a). Reductions in dietary yeast, a significant source of protein in the diet, decrease the growth rate and result in smaller body sizes[34]. When the amount of yeast was reduced by five-fold (1/5Y), the lethality among $Tps1^{MIC}$ mutants increased for both sexes, but this effect was not observed for a three-fold reduction in yeast (1/3Y) (Fig. 5b). Compared with flies fed with the ND, control flies fed with reduced-yeast diets had smaller wings and increased IIV in a dose-dependent manner (Fig. 5c, d). Interestingly, under yeast-poor conditions, $Tps1^{MIC}$ mutants displayed more drastically decreased wing sizes and significantly increased IIV compared with control flies. However, FA did not change in either control or $Tps1^{MIC}$ mutants under yeast-poor conditions (Fig. 5e), indicating that reduced dietary protein levels worsened size deviations between individuals but not within the individuals. The genotype-by-environment interaction demonstrates that Tps1 mutants are sensitive to environmental perturbations.

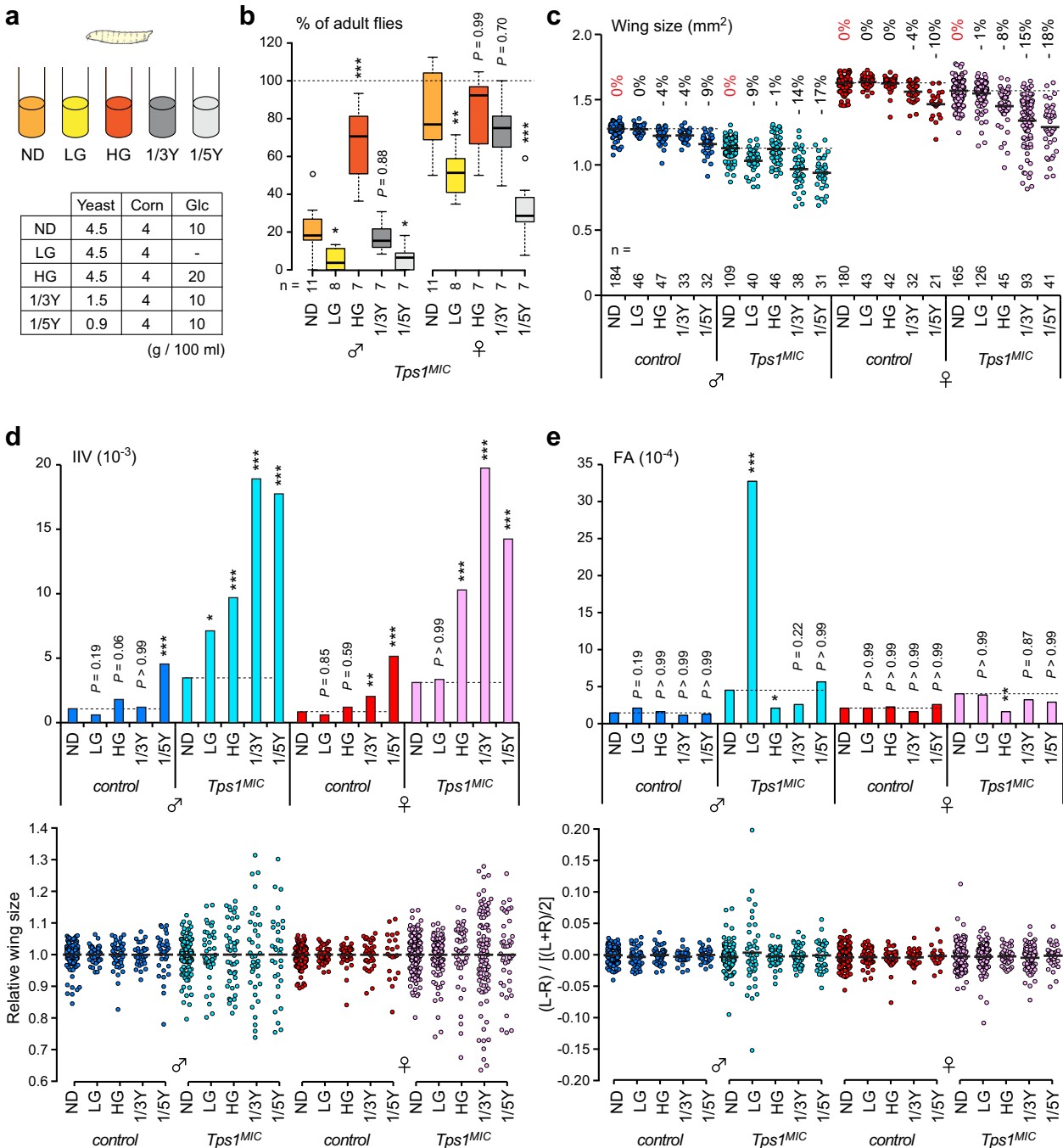

**Fig. 5 Dietary conditions influence developmental homeostasis in *Tps1* mutants. a** Composition of diets used in this study. Corn, cornflour; Glc, glucose. **b** The survival rates of *Tps1^MIC^* mutants for each dietary condition. The percentage of adult flies was determined by the ratio of mutant flies to flies with a balancer chromosome in each vial. **c** Wing sizes for each dietary condition. The percentage differences for the average wing sizes relative to flies grown on a ND are shown above. **d**, **e** Inter-individual variation (IIV) (**d**) and fluctuating asymmetry (FA) (**e**) of wing sizes for each dietary condition. $*p < 0.05$, $**p < 0.01$, $***p < 0.001$; one-way ANOVA with Dunnett's post hoc test (**b**), F-test with Bonferroni correction (**d**, **e**). The numbers of vials (**b**) or flies (**c**–**e**) analyzed are indicated.

We further tested the effects of LG and HG diets. Changing the amount of dietary glucose had limited impacts, if any, on wing size for both control and *Tps1^MIC^* mutant flies when compared with changing the amount of dietary protein (Fig. 5c). We found that control flies did not show changes in IIV or FA when grown on either LG or HG diets, indicating that control flies are mostly resistant to changing levels of dietary glucose with regards to developmental homeostasis. Remarkably, the LG diet drastically

increased FA in *Tps1^MIC^* mutant males but not in females (Fig. 5e). In contrast, the HG diet ameliorated the increase in FA observed in *Tps1^MIC^* mutants grown on the ND (Fig. 5e). Importantly, the changes in FA were well-correlated with changes in survival rate; the LG diet sharply increased and the HG diet rescued the lethality in *Tps1* mutants (Fig. 5b). However, *Tps1^MIC^* mutants showed significantly increased IIV when grown on the HG diet in both sexes (Fig. 5d), indicating that

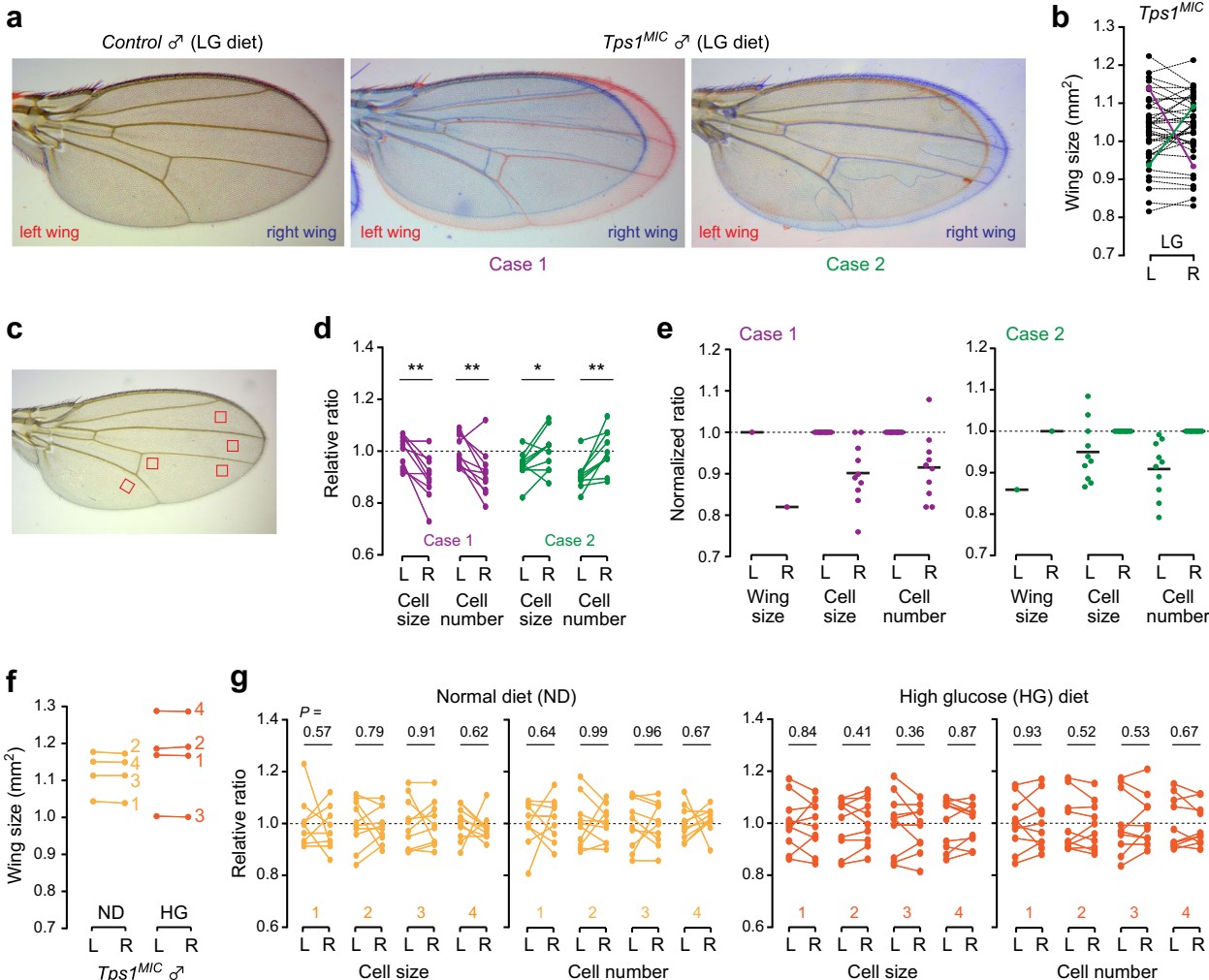

**Fig. 6 Asymmetric size reduction in *Tps1* mutants is caused by changes in both cell size and cell number. a** Merged images of left (red) and right (blue) wings from control and *Tps1*[MIC] mutant males. **b** The size differences between left (L) and right (R) wings for individual flies. The purple and green lines mark two extreme cases. **c** The red squares indicate the regions used for the measurement of cell size. Both dorsal and ventral sides for each region were analyzed: in total, 10 areas in each wing were analyzed. **d** Relative ratios for cell size and cell number between left and right wings in each area. The average values for the larger side (left wing for case 1 and right wing for case 2) were set to 1. **e** Normalized differences in wing size, cell size, and cell number, modified from the data shown in (**d**). **f** The size differences between left and right wings in individual flies. Four randomly selected samples for each dietary condition are shown. **g** Relative ratios for cell size and cell number between the left and right wings of individual flies. *p < 0.05, **p < 0.01; paired two-tailed Student's *t*-test (**d**, **g**).

the HG diet enhances IIV and dampens FA in *Tps1*[MIC] mutants. Collectively, these results strongly support the idea that defects in glucose homeostasis reduce developmental homeostasis in *Tps1* mutants.

**Asymmetric organ size in *Tps1* mutants is caused by changes in both cell size and cell number**. To further investigate the cellular basis of the asymmetry observed in mutant flies, we selected two *Tps1*[MIC] mutant individuals that showed drastic deviations between left and right wings when fed the LG diet (Fig. 6a, b). We determined the wing hair densities of five different regions, on both the dorsal and abdominal sides (10 areas in total) of each wing (Fig. 6c). The total cell numbers in each wing were then estimated based on the cell size values for each area. We found that the small-sized wings had smaller cell sizes and fewer cell numbers than the larger wings (Fig. 6d, e). In contrast, randomly selected *Tps1*[MIC] mutants that showed relatively symmetric wing

sizes displayed comparable values for both cell sizes and cell numbers between left and right wings (Fig. 6f, g). These results indicate that changes in both cell size and cell number produce the asymmetric size deviations in *Tps1*[MIC] mutants.

**Discussion**
Understanding the impacts of genotype-by-environment interactions is fundamentally important for unraveling the mechanisms that drive developmental homeostasis. Several genes have been reported to contribute to the developmental robustness and stability in the genetically tractable model organism *Drosophila*, including microRNAs, heat-shock protein chaperones, the growth regulator *dMyc*, the transcription regulator *CycG*, and the inter-organ signal *dilp8*[7,8,35]. Although blood glucose is an essential fuel source that supplies many energy-demanding events, little is known regarding the impacts of metabolic homeostasis on developmental homeostasis. The genetic manipulation of trehalose

metabolism allowed us to directly evaluate the impacts of altered glucose buffering capacities during development. We propose that trehalose metabolism is a noise-buffering mechanism that confers developmental robustness and stability in response to environmental perturbations and developmental noise.

The adjustment between cellular growth and proliferation contributes to developmental homeostasis by achieving consistent organ sizes[36,37]. A proteome-wide association study has revealed that adult wing size positively correlates with the protein levels of many glycolytic enzymes in wing discs[38], suggesting that larger wing discs require increased glucose metabolism. Consistently, we found that the level of trehalose hydrolysis impacts organ size, although the observed effects are relatively mild. The tissue-specific knockdown of glycolytic genes in eyes and wing imaginal discs have been reported to have little, if any, impacts on normal morphogenesis and organ size[39,40], suggesting that other metabolic pathways compensate for reduced glycolysis during normal development. However, reductions in glycolytic activity can attenuate JNK-induced cell death and TGFβ/Hipk-driven tissue overgrowth in imaginal discs[39,41,42]. Thus, glucose metabolism appears to play protective roles against developmental perturbations.

We found that wild-type flies are widely capable of adapting to various dietary conditions with regards to developmental homeostasis, except for the increase in IIV under poor-yeast diet conditions. Reductions in robustness may be caused by competition between individuals over a limited nutrition source. Despite the increase in IIV, FA remained unchanged in control flies following the nutritional variations examined in this study. These results are consistent with previous reports that variations in temperature strongly affect the IIV, but not the FA, of wing sizes in *Drosophila*[43]. Interestingly, larvae fed with a high-carbohydrate diet display more symmetric sex combs, accompanied by reduced survival rates and slower growth[44]. Moreover, the existence of both independent and shared biological mechanisms for robustness and stability have been demonstrated[9,10,43]. Our results strongly support the view that reductions in developmental robustness do not always coincide with reductions in developmental stability.

In contrast with wild-type flies, *Tps1* mutants are vulnerable to nutritional variations during development. Because *Tps1* mutants display feeding-associated hyperglycemia and fasting hypoglycemia, we suggest that hypoglycemia is the likely cause of reduced developmental stability, whereas hyperglycemia is the likely cause of reduced robustness. In support of this reasoning, asymmetry in the wing appears to develop during the post-feeding developmental stage, including the pupal stage, which must be completed using internal stored resources and, thus, represents a metabolically vulnerable period. Strikingly, most of the stored trehalose during the larval stage is consumed during early metamorphosis[21]. In *Drosophila*, the tubular hearts and aorta, which lie along the dorsal side, function to transport body fluid directionally by pumping hemolymph forward in the body cavity[45,46]. Peristaltic muscular movements during locomotion also facilitate hemolymph movement during the larval period. Both sides of the body are assumed to be exposed to nearly identical concentrations of circulating sugars due to the open circulatory system. However, during pupation, part of the larval heart tube is lost, and the remaining cells must be remodeled into the adult heart tube[47]. The remodeling of the circulatory system and peristaltic movement quiescence during metamorphosis may cause local energy supply fluctuations between sides, leading to asymmetric organ sizes due to differences in cell size and cell number. Moreover, subtle differences in glucose availability between sides could amplify developmental noise or dampen noise-canceling mechanisms by disrupting cellular homeostasis.

We detected a statistically significant reduction in IIV among heterozygous *Tps1* mutant males and a significant reduction in

FA among *sTreh* mutant females compared with genetically matched $w^-$ controls. Because both developmental robustness and stability are polygenic and heterozygosity increases developmental homeostasis[4,7,48], the observed changes might be caused by genetic background differences near the intended locus, due to incomplete backcross. Alternatively, these genotypes may have sex-specific beneficial effects on developmental homeostasis and organ growth for unknown reasons.

We also detected sex-specific unfavorable effects on developmental homeostasis. For example, the increased FA associated with *cTreh* and *dilp8* mutations were only observed in males, but not females. Likewise, the drastic increase in FA observed in *Tps1* mutants fed with the LG diet was only observed in males, although *Tps1* mutants demonstrated increased FA in both sexes under ND conditions. A similarly male-biased increase in FA for morphological traits has been reported in *Drosophila*[10], suggesting that developmental stability in males may be more sensitive to genetic and environmental perturbations than in females. The overall sexual dimorphism in body size is caused by female-specific growth rates and weight loss under the regulation of systemic insulin-like growth factor signaling, the cell-autonomous function of the sex determination gene *Sex-lethal* (*Sxl*), and *dMyc*[49–52]. These female-biased signaling pathways may be involved in the suppression of developmental noise caused by genetic and environmental perturbations. Interestingly, growth impairments, associated with poor glycemic control, are more severe in males than females among children with insulin-dependent diabetes, although the pathophysiological mechanisms underlying these differences are not fully understood[53]. The relationship between sex-specific developmental homeostasis and glucose homeostasis would be an interesting issue for further research.

Trehalose metabolism has been suggested to play protective roles against various stresses, such as cold and desiccation[20,27]. Thus, determining whether the buffering function of trehalose metabolism on developmental homeostasis is generalizable to non-dietary environmental variations will also be interesting to explore. Further insight into the molecular mechanisms that underlie the observed reductions in developmental robustness and stability will reveal how genotypes and environments interact, providing a more comprehensive understanding of developmental homeostasis.

## Methods

**Drosophila strains**. The following *Drosophila melanogaster* strains were used: wild-type strains *Canton S* and *Oregon R*, $w^{1118}$ (used as a control), *UAS-Tps1*, *Tps1^MIC* (described previously[21]), *Tps1-full* genomic rescue construct, *Treh^MIC*, *Treh^s1* (described previously[27]), *UAS-sTreh-Flag*, *UAS-cTreh-Flag*, *GlyS^8*, and *GlyP^3–13* (described previously[28]). *Tps1-RNAi* (4104R-6) and *Treh-RNAi* (9364R-1) were obtained from the National Institute of Genetics (NIG) *Drosophila* Stock Center. The following stocks were obtained from the Bloomington *Drosophila* Stock Center (BDSC): *Mi{MIC}dilp8^MI00727* (33079), *GlyS-RNAi* (34930), *GlyP-RNAi* (33634), *mCherry-RNAi* (35787), *UAS-Dcr-2*, *en-Gal4*, *UAS-2xEGFP* (25752). The X chromosome of the *dilp8^MI00727* mutants was replaced with that of $w^-$, and the mutants were used in the $w^-$ background.

**Fly diets**. The animals were reared on fly food (normal diet, ND) that contained 8 g agar, 100 g glucose, 45 g dry yeast, 40 g cornflour, 4 ml propionic acid, and 0.45 g butylparaben (in ethanol) per liter (1× recipe). Low/high-glucose and low-yeast diets were prepared, as shown in Fig. 5a. For the acute high-glucose challenge and starvation experiments, shown in Fig. 1d, mid-third instar larvae (approximately 24 h after the second ecdysis) grown on ND or LG diet were transferred to vials containing either 20% glucose and 0.8% agar in $H_2O$ or 0.8% agar in $H_2O$. No yeast paste was added to the fly tubes in any experiments. All the experiments were conducted under non-crowded conditions at 25 °C.

**Measurement of protein, TAG, and sugar levels**. Measurements of protein, trehalose, glucose, glycogen, and TAG levels in whole larvae were performed as described previously[28,54].

In brief, larvae were rinsed several times with PBS to remove all traces of food. Frozen samples in tubes were homogenized using a pellet pestle in 100 μl of cold PBS containing 0.1% TritonX-100, immediately heat inactivated at 80 °C for 10 min, and then cooled to room temperature (RT). Samples were further crushed to obtain uniform homogenates with $1\times \phi 3$ mm zirconia beads using an automill (Tokken Inc.) at 41.6 Hz for 2 min. The homogenate samples were used to determine the TAG and glycogen levels, and the cleared samples after centrifugation at $12,000 \times$ rpm $(13,000 \times g)$ at RT for 10 min were used to determine the trehalose and glucose levels. Ten μl of the homogenate was mixed with 10 μl of a triglyceride reagent (Sigma-Aldrich), incubated at 37 °C for more than 30 min, and then cleared by centrifugation. Five μl of the supernatant was used for the measurement of TAG by free glycerol reagent (Sigma-Aldrich). A triolein equivalent glycerol standard (Sigma-Aldrich) was used as the standard. Five μl of the sample was incubated with PBS containing amyloglucosidase (Roche) or bacterially produced recombinant His-tagged cTreh at 37 °C overnight. Ten μl of the sample was incubated with PBS without enzymes in parallel for the determination of glucose levels. The reaction was carried out in a 15 μl assay mixture. The amount of glucose was determined by a glucose assay kit (Sigma-Aldrich). A serial dilution of glucose was used as standard. The trehalose and glycogen concentrations for each sample were determined by subtracting the values of free glucose in the untreated samples. The amounts of TAG, trehalose, glycogen, and glucose were normalized to the total protein level as described below.

For determination of total protein levels in larval samples, the homogenate samples were mixed with two volumes of 0.2 N NaOH, vortexed for 10 min at RT, heated at 95 °C for 15 min to solubilize proteins. The cleared samples after centrifugation at 12,000 rpm $(13,000 \times g)$ for 10 min at RT were used to quantify protein using a BCA protein assay kit (Thermo).

**Measurement of circulating sugar levels by LC-MS/MS.** Staged larvae were collected, rinsed with PBS, and dried on tissue paper. The cuticle was carefully torn to release the hemolymph onto a parafilm membrane. One microliter hemolymph was collected using a micropipette, mixed with 300 μl cold methanol, and stored at −80 °C until extraction. The samples were mixed with 300 μl $H_2O$ and 100 μl $CHCl_3$, vortexed for 20 min at RT, and centrifuged at 15,000 rpm $(20,000 \times g)$ for 15 min at 4 °C. The supernatant (550 μl) was transferred to a new 1.5-ml tube, dried in a vacuum concentrator, re-dissolved in 1 mM ammonium bicarbonate, and analyzed by LC-MS/MS.

Chromatographic separation and mass spectrometric analyses were performed essentially as described previously[27,55]. Chromatographic separation was performed on an ACQUITY BEH Amide column (100 mm × 2.1 mm, 1.7 μm particles, Waters) in combination with a VanGuard precolum (5 mm × 2.1 mm, 1.7 μm particles) using an Acquity UPLC H-Class System (Waters). Elution was performed at 30 °C under isocratic conditions (0.3 mL/min, 70% acetonitrile, and 30% 10 mM ammonium bicarbonate, pH 10.0). The mass spectrometric analysis was performed using a Xevo TQD triple quadrupole mass spectrometer (Waters) coupled with an electrospray ionization source in the negative ion mode. The multiple reaction monitoring transitions were as follows: $m/z$ 341.2–89.0 for trehalose, $m/z$ 179.1–89.0 for glucose and fructose, and $m/z$ 181.1–89.0 for sorbitol. Analytical conditions were optimized using standard solutions. Sample concentrations were calculated from the standard curve obtained from a serial dilution of the standard solution.

**Wing morphometric analysis.** Adult flies were photographed under a Zeiss Stemi 2000-C stereomicroscope (Zeiss) equipped with a Canon PowerShot G15 digital camera (Canon). Adult flies with indicated genotypes and dietary conditions were fixed in isopropanol in a 1.5-ml tube. Left/right wings were removed in 1× PBS under a Zeiss Stemi 2000 stereomicroscope (Zeiss) and mounted dorsal side up on a glass slide using 25% glycerol/25% isopropanol. Digital images were acquired with a Zeiss Primo Star stereomicroscope equipped with AxioCam ERc (Zeiss). The wing area, as indicated in Fig. 2c, was manually traced and measured using either ImageJ or AxioVision (Zeiss) software.

The wing sizes shown in Figs. 4c and 5c indicate the average sizes between the right and left wings for individuals. Inter-individual variation (IIV) for wing size was expressed as the variance among the average individual wing sizes within populations. Fluctuating asymmetry (FA) was estimated using FA index 6, as described previously[3], which is based on a signed difference between sides and scaled by average individual size. Namely, FA is the variance expressed as $(L - R)/[(L + R)/2]$, where L and R are the left and right wings, respectively. The presence of directional asymmetry (DA) was investigated by one-sample Student's $t$-test for each data sample $[Ho = \text{mean}(L - R) = 0]$. We detected statistical significance $(p < 0.01)$ for several genotypes and dietary conditions, suggesting the presence of DA. Of note, the FA index 6 remains unaffected by DA[3]. The biological significance and molecular mechanisms underlying DA remain unknown. Experiments were repeated with at least two independently reared populations, and all data were combined. IIV and FA were evaluated by an $f$-test with a Bonferroni correction for multiple comparisons.

To evaluate the measurement error, we performed two measurements per side for each sample during our first experiment. The average variances for the measurement error were 16.6 % (control flies) and 13.4% ($Tps1^{MIC}$ mutant flies) of

the respective variance calculated for FA ($p < 0.0001$ by the $f$-test in all cases). We also determined whether the between-sides variations were larger than the measurement error using a two-way analysis of variance (ANOVA). The interaction term was significant with side and individual as factors ($p = 0.0001$ in control flies and $p = 0.0015$ in the $Tps1^{MIC}$ mutant flies). Thus, the contribution of measurement error to overall variation was relatively small; therefore, we decided to use one measurement per side to calculate FA. All measurements per data set were performed by the same person.

The measurements of cell size and cell number in the adult wing were performed as described previously[56] with minor modifications. The dorsal and ventral sides of the five defined regions, as indicated in Fig. 6c, were acquired separately with a Zeiss Primo Star microscope equipped with AxioCam ERc (Zeiss). Cell density was manually analyzed by counting the number of wing hairs in a 0.01-mm² area, and cell size and approximate cell number per wing were calculated. For data shown in Fig. 2b, a single region in the third posterior cell was analyzed per wing. For wing compartment analysis, as shown in Fig. 2d, the wing area and the posterior region (the area between L4 and the wing margin) were manually measured in ImageJ. The anterior region was calculated by subtracting the posterior region from the wing area.

**Mating competition assay.** Male flies were collected within 12 h of eclosion and were maintained on ND for 3 days. Two males (a control and a $Tps1$ mutant) and a virgin female (*Oregon R* or *Canton S*) were placed into a vial together for mating and oviposition. Parents were discarded 3 days later, and the subsequent F1 males were collected for genotyping to identify the parent male. In brief, genomic DNA was extracted by homogenizing a single fly in 50 μl of buffer [10 mM Tris/HCl (pH 8.0), 1 mM EDTA, 25 mM NaCl, 2 μg/ml Proteinase K (Roche)] using a pellet pestle. The samples were incubated at 37 °C for 30 min, heat-inactivated at 95 °C for 10 min, and centrifuged at 15,000 rpm $(20,000 \times g)$ for 10 min at RT. The supernatant was subjected to PCR amplification to determine the presence of the *Minos* insertion. Experiments were repeated twice using independently reared populations, and all data were combined.

**Fertility test.** Male flies were collected within 12 h of eclosion and maintained on ND for 3–7 days. Single males ($Tps1^{MIC}$ heterozygote or homozygote) and three virgin females ($w^-$) were placed in a vial for mating and were transferred to fresh vials every day until 3 days after mating. Male fertility was assessed by the presence of larvae. The average number of progenies was determined by counting adult flies from three consecutive vials. Experiments were repeated twice using independently reared populations, and all data were combined.

**Statistics and reproducibility.** The experiments were replicated at least twice using independently rared populations to ensure reproducibility. The experiments were not randomized, and the investigators were not blinded to fly genotypes during experiments. Sample sizes for each experiment are indicated in the figures and figure legends. Statistical tests were performed in Microsoft Excel and GraphPad Prism 7 software. The statistical tests used are described in the figure legends. Statistical significance was presented as follows: $*p < 0.05$, $**p < 0.01$, $***p < 0.001$.

**Reporting summary.** Further information on research design is available in the Nature Research Reporting Summary linked to this article.

## Data availability

Correspondence and requests for materials should be addressed to T.N. (email: takashi.nishimura@riken.jp). All data supporting the findings of this study are included in this article. The source data presented in the main figures are provided in Supplementary Data 1.

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

## Acknowledgements

We thank TRiP at Harvard Medical School, the Bloomington *Drosophila* Stock Center, and the National Institute of Genetics *Drosophila* Stock Center for fly stocks. We thank the members of the fly laboratories at RIKEN BDR for their valuable support and discussions. We thank Yuki Yamauchi for support performing wing morphometric analyses. We thank Ken-ichi Hironaka for sharing illustrations and stimulating discussions. We also thank Kazuo Takahashi for discussions and helpful comments on the manuscript. This work was supported, in part, by the Japan Society for the Promotion of Science (KAKENHI grants JP17K19433 and JP17H03658 to T.N.).

## Author contributions

R.M. and T.N. designed and performed the experiments. R.M. and T.N. performed the data analysis. T.N. wrote the paper. All authors discussed and reviewed the paper.

## Competing interests

The authors declare no competing interests.
