## [Peer Review File · Communications Biology]

Reviewers' comments:

Reviewer #1 (Remarks to the Author):

The study of Matsushita and Nishimura addresses the importance of trehalose metabolism to *Drosophila* glucose homeostasis, tissue growth as well as developmental robustness and stability. The authors have observed that hypomorphic mutants of trehalose biosynthesis enzyme Tps1 show strong reduction of circulating trehalose as well as dietary glucose-dependent elevation of glucose, fructose and sorbitol. This is consistent with the conclusion that trehalose serves as an important glucose sink in *Drosophila*. Interestingly the authors also observe that trehalose catabolism is needed for cell growth in the wing, and that impaired trehalose biosynthesis leads to increased inter-individual variation and fluctuating asymmetry of wing size. Finally, the authors provide evidence that paternal trehalose metabolism affects fluctuating asymmetry in the offspring. In most parts, the data is of good quality, experiments are properly controlled, conducted and documented and the conclusions are in line with the data. The findings increase our understanding on the homeostatic role of insect trehalose metabolism and on the contribution of metabolism on developmental robustness and stability. Consequently, this study will be of interest to scientists on related fields.

I have two main concerns that should be addressed prior to publication.

1. All the experiments on Tps1 have been done with a single mutant genotype, a hypomorphic mutant Tps1^{MIC}. The authors should confirm the main findings (especially the influence of Tps1 on inter-individual variation and fluctuating asymmetry) by an independent Tps1 mutant or RNAi, or alternatively by performing genetic rescue of Tps1^{MIC} by transgenic Tps1.

2. I am doubtful about the conclusions regarding the effects of the paternal metabolic state (Fig.7). This conclusion is based on a borderline significant difference (P-value is 0.046) on fluctuating asymmetry between offspring of males reared on ND and HG diets. Looking at the data points of HG and ND one can observe that 37 out of 40 data points of HG flies are distributed similarly to the 40 data points of ND, while three HG data points deviate clearly from the rest of the data. Therefore, I conclude that the current dataset does not sufficiently support the conclusion about the paternal effect. The authors should either provide an independent line of evidence to strengthen their conclusion or omit the dataset from the manuscript.

Reviewer #2 (Remarks to the Author):

This is a good contribution to the field. There is some typing error like page 10 2nd para Occurr->occur. It should be corrected. What happened to eye imaginal disc? The authors should show if there is no change in the eye.

Reviewer #3 (Remarks to the Author):

The authors investigated the interplay between glucose homeostasis and developmental homeostasis. They focused on the impact of trehalose metabolism on the metabolism and developmental homeostasis of *D. melanogaster*. They found that trehalose metabolism is critical for robustness and developmental stability of the adult wing size. They stated that trehalose-6-phosphate synthase 1 (Tps1) mutants showed feeding-associated hyperglycemia and fasting hypoglycemia. They also reported that Tps1 mutants increased the among-individual and within-individual variations in the

wing size.

Overall, this paper describes potentially interesting findings, however, the way the data is presented makes it hard for readers of *Communications Biology* to understand the significance of this study. Most importantly, I suspect that all the information throughout the manuscript is indeed necessary. The manuscript will be improved and of interest to the readers particularly to researchers in the fields of developmental biology and metabolism, once the issue is resolved.

Major comments:

1. The authors should carefully reconsider which data in the main figures are invaluable. For example, Figure 2e and Figure 4a-c and their related descriptions are not essential for their main findings and it might be better to move them to supplementary figures.
2. It would be useful to present representative images of the wings before showing the quantification of IIV and FA in Figures 3, 5, and 7.
3. In the Discussion section, the authors state that "Interestingly, maternal obesity and diabetes have been associated with increased levels of FA in human and rhesus monkey offspring. These findings suggest that hyperglycemia-induced glucotoxicity can negatively impact offspring symmetry in diverse species." This information provides an important background to the study and should be moved to the Introduction section.

Minor comments:

1. Dot plots in Figures 3a, 3b, 4a, 4b, 5d, 5e, 7b, 7c need item names.
2. It is better to delete "for the first time" in p5.
3. Correct a typographical error in "trehalose synthesis enzye (Tps1)" in p2.
4. The author used a word "proper" in p2, p4, and p9. I think it might be better to use "appropriate" instead.
5. "than the larger wings (Fig. 6c, d)" in p 11 and "wings (Fig. 6e, f)" in p12 should be "than the larger wings (Fig. 6d, e)" and "wings (Fig. 6f, g)", respectively.

Reviewers' comments:

Reviewer #1 (Remarks to the Author):

The study of Matsushita and Nishimura addresses the importance of trehalose metabolism to *Drosophila* glucose homeostasis, tissue growth as well as developmental robustness and stability. The authors have observed that hypomorphic mutants of trehalose biosynthesis enzyme *Tps1* show strong reduction of circulating trehalose as well as dietary glucose-dependent elevation of glucose, fructose and sorbitol. This is consistent with the conclusion that trehalose serves as an important glucose sink in *Drosophila*. Interestingly the authors also observe that trehalose catabolism is needed for cell growth in the wing, and that impaired trehalose biosynthesis leads to increased inter-individual variation and fluctuating asymmetry of wing size. Finally, the authors provide evidence that paternal trehalose metabolism affects fluctuating asymmetry in the offspring. In most parts, the data is of good quality, experiments are properly controlled, conducted and documented and the conclusions are in line with the data. The findings increase our understanding on the homeostatic role of insect trehalose metabolism and on the contribution of metabolism on developmental robustness and stability. Consequently, this study will be of interest to scientists on related fields.

I have two main concerns that should be addressed prior to publication.

1. All the experiments on *Tps1* have been done with a single mutant genotype, a hypomorphic mutant *Tps1^{MIC}*. The authors should confirm the main findings (especially the influence of *Tps1* on inter-individual variation and fluctuating asymmetry) by an independent *Tps1* mutant or RNAi, or alternatively by performing genetic rescue of *Tps1^{MIC}* by transgenic *Tps1*.

Thank you for your comment. We used a *Tps1* genomic rescue construct that we had previously generated (Yoshida et al., 2016) to confirm our primary findings and found that the use of a genomic rescue construct carrying a single wild-type *Tps1* allele fully suppressed the observed increases in inter-individual variation and fluctuating asymmetry caused by the *Tps1^{MIC}* mutations (Figure 3a, b), which strongly supports our primary findings.

None of the independent *Tps1* mutants survived to the adult stage. The knockdown of *Tps1* causes 100% lethality during metamorphosis and soon after eclosion (Matsuda et al., 2015), precluding adult wing analyses.

2. I am doubtful about the conclusions regarding the effects of the paternal metabolic state (Fig.7). This conclusion is based on a borderline significant difference (P-value is 0.046) on fluctuating asymmetry between offspring of males reared on ND and HG diets. Looking at the data points of HG and ND one can observe that 37 out of 40 data points of HG flies are distributed similarly to the 40 data points of ND, while three HG data points deviate clearly from the rest of the data. Therefore, I conclude that the current dataset does not sufficiently support the conclusion about the paternal effect. The authors should either provide an independent line of evidence to strengthen their conclusion or omit the dataset from the manuscript.

We agree with the reviewer's comment and have opted to omit this dataset from the manuscript. We will attempt to obtain further molecular genetic evidence before reporting these results or making any conclusions.

Reviewer #2 (Remarks to the Author):

This is a good contribution to the field. There is some typing error like page 10 2nd para Occurr->occur. It should be corrected. What happened to eye imaginal disc? The authors should show if there is no change in the eye.

Thank you for your comment. We have corrected the typo on page 10. Regarding the eye, we did not observe any morphological defects in any of the examined genotypes when using a stereomicroscope. In this manuscript, we focused on the wing as a readout of developmental robustness and stability because the adult wing is a suitable organ for precisely monitoring size differences among varying conditions.

Reviewer #3 (Remarks to the Author):

The authors investigated the interplay between glucose homeostasis and developmental homeostasis. They focused on the impact of trehalose metabolism on the metabolism and developmental homeostasis of *D. melanogaster*. They found that trehalose metabolism is critical for robustness and developmental stability of the adult wing size.

They stated that trehalose-6-phosphate synthase 1 (*Tps1*) mutants showed feeding-associated hyperglycemia and fasting hypoglycemia. They also reported that *Tps1* mutants increased the among-individual and within-individual variations in the wing size.

Overall, this paper describes potentially interesting findings, however, the way the data is presented makes it hard for readers of *Communications Biology* to understand the significance of this study. Most importantly, I suspect that all the information throughout the manuscript is indeed necessary. The manuscript will be improved and of interest to the readers particularly to researchers in the fields of developmental biology and metabolism, once the issue is resolved.

Major comments:

1. The authors should carefully reconsider which data in the main figures are invaluable. For example, Figure 2e and Figure 4a-c and their related descriptions are not essential for their main findings and it might be better to move them to supplementary figures.

Thank you for your comment. We agree with the reviewer's comment and have deleted Figure 2e (glycogen distribution in wing imaginal discs) because the knockdown of glycogen metabolism genes had no effects on adult wings. However, we believe that Figures 4a-c should be included in the main text. This figure is important for demonstrating the specificity of the *Tps1* mutant phenotype and the phenotypic differences between the *Tps1* mutants and previously reported mutants.

2. It would be useful to present representative images of the wings before showing the quantification of IIV and FA in Figures 3, 5, and 7.

Although the reviewer has asked that we present representative wing images, we would like to emphasize that the degrees of both inter-individual variation and fluctuating asymmetry are judged according to within-population deviations and that these values are strongly affected by the presence of outliers within a sample population. Therefore, in populations with strong inter-individual variation and fluctuating asymmetry, such as the one described in our study, determining which images are “representative” of the population can be quite difficult. Instead, we have included dot plots in all figures to demonstrate the size variations within each population.

In the revised manuscript, we showed examples of left-right wing size differences in *Tps1* and *Treh* mutants (Figure 3c), related to Figure 3b. We think these images would be useful for comparisons against the phenotypes shown in Figures 4 and 5.

3. In the Discussion section, the authors state that “Interestingly, maternal obesity and diabetes have been associated with increased levels of FA in human and rhesus monkey offspring. These findings suggest that hyperglycemia-induced glucotoxicity can negatively impact offspring symmetry in diverse species.” This information provides an important background to the study and should be moved to the Introduction section.

According to a suggestion made by Reviewer #1, we have deleted this section from the revised manuscript.

Minor comments:

1. Dot plots in Figures 3a, 3b, 4a, 4b, 5d, 5e, 7b, 7c need item names.

We have added genotype labeling to these panels.

2. It is better to delete “for the first time” in p5.

We have deleted this phrase, as suggested.

3. Correct a typographical error in “trehalose synthesis enzye (*Tps1*)” in p2.

We have corrected this typo.

4. The author used a word “proper” in p2, p4, and p9. I think it might be better to use “appropriate” instead.

We have changed the wording here, as suggested.

5. “than the larger wings (Fig. 6c, d)” in p 11 and “wings (Fig. 6e, f)” in p12 should be “than the larger wings (Fig. 6d, e)” and “wings (Fig. 6f, g)”, respectively.

We have fixed this mistake.

REVIEWERS' COMMENTS:

Reviewer #1 (Remarks to the Author):

The authors have properly addressed my previous concerns. I have no additional requests. I recommend to publish the current version of the manuscript.

Reviewer #3 (Remarks to the Author):

The authors have addressed all of my comments.
It has become acceptable.